# A New Class of Autopoietic and Cognitive Machines

**Rao Mikkilineni** 

Ageno School of Business, Golden Gate University, San Francisco, CA 94105, USA; rmikkilineni@ggu.edu

**Abstract:** Making computing machines mimic living organisms has captured the imagination of many since the dawn of digital computers. However, today's artificial intelligence technologies fall short of replicating even the basic autopoietic and cognitive behaviors found in primitive biological systems. According to Charles Darwin, the difference in mind between humans and higher animals, great as it is, certainly is one of degree and not of kind. Autopoiesis refers to the behavior of a system that replicates itself and maintains identity and stability while facing fluctuations caused by external influences. Cognitive behaviors model the system's state, sense internal and external changes, analyze, predict and take action to mitigate any risk to its functional fulfillment. How did intelligence evolve? what is the relationship between the mind and body? Answers to these questions should guide us to infuse autopoietic and cognitive behaviors into digital machines. In this paper, we show how to use the structural machine to build a cognitive reasoning system that integrates the knowledge from various digital symbolic and sub-symbolic computations. This approach is analogous to how the neocortex repurposed the reptilian brain and paves the path for digital machines to mimic living organisms using an integrated knowledge representation from different sources.

**Keywords:** cognition; computing models; deep learning; autopoiesis; knowledge structures structural machines; autopoietic machines

## 1. Introduction

To make computing machines mimic living organisms, first, we must understand the unique features of the living organisms that make them sentient, resilient, and intelligent. Physical and mental structures which transform information and knowledge are the essential ingredients of all living organisms. Our knowledge about these structures comes from genomics [1], neuroscience [2,3], cognitive science [4], and the studies of artificial intelligence [5,6]. The references cited provide a compelling picture of the information processing structures used by the living organisms and their role in managing the "life" processes with varying degrees of sentience, resilience, and intelligence.

In most living organisms, genes encode the life processes and pass them on from the survivor to the successor. The genetic knowledge structures include executable workflows and control processes that describe stable patterns of the living organism. These processes are designed to optimally utilize the resources available to assure the organism's creation and safekeeping when interacting with its environment. Creation involves the processes that use knowledge to transform matter and energy. The system with "self" awareness is assembled using physical structures with several constituent components. Safekeeping involves the ability to dynamically monitor and control an organism's behavior along with its interactions with its environment using genetic descriptions. Intelligent systems augment inherited knowledge through genes with cognitive processes embedded in the nervous systems and neural networks. The system uses its components to gain information through its sensory components and converts it to knowledge using its neural networks. The neurons which fire together are wired together to capture the knowledge about events that caused the firing, and the neurons that are wired together fire together to exhibit autopoietic and cognitive behaviors.

Both autopoiesis and cognition are capabilities exploited by living organisms. They are the essence of an organism's sentient, resilient, and intelligent behaviors that contribute towards managing its stability, safety, and sustenance. Autopoiesis refers to the behavior of a system that replicates itself and maintains identity and stability while its components face fluctuations caused by external influences. Autopoiesis enables them to use the specification in their genomes to instantiate themselves using matter and energy transformations. They reproduce, replicate, and manage their stability using cognitive processes. Cognition allows them to process information into knowledge and use it to manage its interactions between various constituent parts within the system and its interaction with the environment. Cognition uses various mechanisms to gather information from various sources, convert it into knowledge, develop a history through memorizing the transactions, and identify new associations through their analysis. Organisms have developed various forms of cognition. According to Burgin [private communication],

A process is:

- embedded if it goes in some physical or mental system; for example, the process of walking in the street is embedded in this street but not embodied in it.
- embodied if it goes in the system that maintains it; for example, the process of computation going in a computer.
- enacted if it is initiated by the system where it goes, by a system involved in the process, or by another system.
- elevated if there is a hierarchy of processes and the process goes on higher levels of this hierarchy; for example, the hierarchy of inductive Turing machines and processes within them.
- extended if it moves outside the system in which it started or if it goes beyond some boundary in time.
- efficient if it produces high-quality results being provided with sensible resources.
- endogenous if it has an internal cause or origin.

In short, the living organism's computing models, consisting of complex multi-layer networks of genes combined with neural network processing, enable the formulation of descriptions and execution of workflow components having not only the content of how to accomplish a task but also providing the context, constraints, control, and communication to assure systemic coordination to accomplish the overall purpose of the system embedded in the genome. Various constituent structures process information and convert it into knowledge which is integrated and used by a higher level of cognition known as elevated cognition.

Intelligent systems have also developed internal and external communication structures that allow sentient behavior (the ability to sense and react). Computing (the ability to transform information obtained through the senses, create and process knowledge structures capturing the dynamics), communication (the ability to pass information within its components and with external systems) and cognition (the ability to create and execute processes that sense and react to changing circumstances) are essential ingredients of intelligence that provide sentience and resilience (the ability to know and adapt appropriately to changing circumstances).

Biological structures are described as complex adaptive systems (CAS) composed of many interrelated and interacting components (made up of components that exploit the properties of atoms, molecules, compounds, etc., to create the composed structures). CAS [7] exhibits self-organization, non-linearity, the transition between states of order and chaos, and emergence. The system often exhibits behavior that is difficult to explain through an analysis of the system's constituent parts. Such behavior is called emergent. CAS are complex systems that can adapt to their environment through an evolution-like process and are isomorphic to networks (nodes executing specific functions based on local knowledge and communicating information using links connecting the edges). The system evolves into a complex multi-layer network, and the functions of the nodes and the composed structure define the global behavior of the system as a whole. Sentience,

resilience, and intelligence are the result of these structural transformations and dynamics exhibiting autopoietic and cognitive behaviors.

If digital machines were to mimic living organisms, we must endow them with the ability to exhibit autopoietic and cognitive behaviors. Fortunately, the general theory of information (GTI) [8,9], our understanding of structural reality [10], and various tools derived from them [11–15] provide a new approach to not only model autopoietic and cognitive behaviors in living organisms but also provide a new method to infuse them into digital automata.

The thesis of this paper is that if digital machines were to mimic living organisms' sentient, resilient, and intelligent behaviors, then they must be infused with autopoietic and cognitive behaviors. Current information-processing structures with symbolic computing (based on John von Neumann's stored program implementation of the Turing machine) and deep learning (based on algorithms that mimic neural networks) fall short [16–19] of mimicking the autopoietic and cognitive behaviors of living organisms. Software applications lack self-management and depend on external entities to find resources, deploy, configure, monitor, and manage them. Current deep learning algorithms such as CNN (convolution neural networks), RNN (reinforced neural networks), etc., while they are very successful in providing knowledge insights from information gathered and represented in the form of symbolic data structures, do not provide the basic ingredients that are required to integrate various knowledge insights from multiple sets of inputs from different sources. In short, they are unable to provide a common knowledge representation from different neural networks processing different data sets just as the mammalian neocortex and the reptilian cortical columns do, as we shall see later. The required ingredients are:

1.  A systemic view of the myriad relationships among the knowledge components inferred from different sources (equivalent to the sense of "self" and its structural relationships with the entities with which they interact using 7e cognition),
2.  A common knowledge representation to encapsulate the dynamic interactions and component behavioral evolution as events influence changes in the system, and
3.  A sense of history and the best practices to reason at a higher level with shared knowledge from multiple inputs from multiple components to optimize global behavior to address fluctuations that impact the stability, safety, and sustenance of the system (elevated cognition).

In essence, current symbolic and sub-symbolic components constitute a CAS and, left to themselves, they are subject to the properties of non-deterministic emergence in the face of large fluctuations in the system component interactions. Autopoietic and elevated cognition (known as super-symbolic computation [15]) provides the mechanisms to maintain stability, safety and optimize the system's global behavior.

In this paper, we use GTI to discuss the evolution of sentience, resilience, and intelligence in living organisms. We examine information processing structures and discuss a theoretical model providing their essential characteristics such as autopoiesis and cognition. The model is derived from GTI and described in [8–15]. This model allows us to design ways to infuse autopoietic and cognitive behaviors into digital information processing structures built using digital automata. Section 2 describes the lessons from studying the evolution of autopoietic and cognitive behaviors in living organisms. In Section 3, we present a theoretical model based on the general theory of information and the theory of structures to replicate the structures exhibiting the autopoietic and cognitive behaviors. In Section 4, we describe a new approach to integrating knowledge from multiple sources such as symbolic and sub-symbolic computations with a common knowledge representation, and provide model-based reasoning to mitigate risk. In Section 5, we conclude with some observations on this approach and its impact on information technology's current and future state.

## 2. Evolution of Sentience, Resilience, and Intelligence in Living Organisms

### 2.1. Intelligence and Natural Evolution

An important question is—how did living organisms evolve from being mere physical and chemical structures to develop the complex behaviors of autopoiesis and cognition we observe in all living beings with varying degrees of sentience, resilience, and intelligence? Experts tell us that "In the microbial world, decisions are made by monitoring the current state of the system, by processing this information and by taking action with the ability to take into account several factors such as recent history, the likely future conditions and the cost and benefit of making a particular decision. At the population level, microbes are also capable of hedging their bets, by having individuals of an isogenic population in different states even when experiencing the same environmental conditions, and they are also able to make collective decisions that cause the entire population to respond in a particular way. Microbes can make decisions based on different criteria of information, and also perform the decision-making using different mechanisms, utilizing different types of molecular networks [20] p. 5."

As Darwin said in the conclusion of his *'long argument'*, "And as natural selection works solely by and for the good of each being, all corporeal and mental endowments will tend to progress towards perfection [21] p. 506". He also said "Natura non facit saltum" or nature does not make jumps [21] p. 489.

Indeed, it seems that nature chose "punctuated equilibria" for evolution with increasing levels of sentience resilience and intelligence. According to Westerhoff, "Microbes exhibit similar characteristics of intelligence as higher organisms and humans, such as decision-making, robust adaptation, association and anticipation, self-awareness and problem-solving capabilities [20] p. 6." Living organisms persist under complex interactions among many components organized into dynamic, environment-responsive networks that span multiple scales and dimensions. These studies show that the evolution of biological systems from the underlying physical and chemical structures was a gradual transformation of independent component structures interacting with each other and behaving like a complex adaptive system. The system's evolution based on individual component structure and function and their interactions with each other and the external environment is the result of emergent properties of a complex adaptive system. As fluctuations in their interactions and the scale of the components increased, the emergent property allowed the formation of complex multi-layer networks with behaviors that were different from any of the individual components.

Let us go back to the question, what did the living organisms evolve from and how? Our current scientific theories point to a structural evolution from physical and chemical components aided by random fluctuations in their interactions among themselves and with their environment. Function, structure, and fluctuations play a key role in the evolution of physical and chemical structures obeying the laws of transformation governing matter and energy. The laws of thermodynamics influence the microscopic and macroscopic behaviors of these structures. According to the first law of thermodynamics, if the energy of the system consisting of structures that are interacting with each other is conserved, it would reach equilibrium and the structures would tend to be stable. The second law of thermodynamics states that if a closed system is left to itself, it tends to increase disorder and entropy, which is a measure of the disorder. However, if the system can exchange energy with the environment outside, it can increase its order by decreasing entropy inside and transferring it to the outside. This allows the structures to use energy from outside and form more complex structures with lower entropy or higher order.

These concepts have been the foundation for the theories of phase transitions in physics (as Prigogine mentioned in his Nobel lecture, "nonequilibrium may be a source of order" [22,23] p. 1), and the theory of complex adaptive systems which has been applied to understand the economic behaviors of groups engaged in commerce [7] ("In the complex adaptive system of the economy, understanding the micro-level behaviors of individuals is essential to understanding how the system as a whole behaves" [7] p. 194). Living beings

have, through evolution and natural selection, perfected the art of increasing order inside by exchanging energy and entropy with the outside.

Five "Big Ideas" that distinguish complex adaptive systems are [7] the following:

4.  They are open, dynamic, and nonlinear systems, which are far from equilibrium.
5.  Individual components interact with each other and their environment without a global view of information. Local dynamics and behaviors are completely dependent on energy and entropy considerations and the energy exchange.
6.  The system dynamics often, results in the formation of networks where each component interacts with other components and could be modeled as nodes with specific functional behavior (exploiting matter and energy transformations), exchanging information with other components based on energy and entropy considerations; networks of relationships change over time.
7.  As fluctuations cause local variations, their scale and magnitude determine the degree of disequilibrium of the system and the system exhibits emergence where the global behavior of the system is completely unpredictable based on the behaviors of individual components.
8.  The evolutionary process of differentiation, selection, and amplification provides the system with novelty and is responsible for its growth in order and complexity.

It seems that living organisms have evolved from the soup of physical and chemical structures [20] in three different phases, where the characteristics of the evolving systems are different. In all phases, evolution involves complex multi-layer networks. According to Westerhoff et al., [20] "Biological networks constitute a type of information and communication technology (ICT): they receive information from the outside and inside of cells, integrate and interpret this information, and then activate a response. Biological networks enable molecules within cells, and even cells themselves, to communicate with each other and their environment. [20] p. 1."

### 2.1.1. Phase 1: Metabolic Networks

In a cell or microorganism, the processes that generate matter, energy, information transfer, and the cell fate specification are seamlessly integrated through a complex network of various cellular constituents and reactions [24]. Biologists and theoreticians [25] have analyzed several organisms and shown that, despite significant variances in their constituents and pathways, metabolic networks display the same topologic scaling properties demonstrating striking similarities to the inherent organization of complex non-biological systems. They conclude that metabolic organization is not only identical for all organisms, but complies with the design principles of robust and error-tolerant scale-free networks, and may represent a common blueprint for the large-scale organization of interactions among all cellular constituents. The scale-free networks are characterized by their degree of distribution which is a Poisson distribution, whereas a random network is characterized by a Gaussian distribution. The Poisson distribution is used to describe the distribution of rare events such as metabolic network formation in a large population. The degree of a node in a network is the number of connections a node has to other nodes, and the degree distribution is the probability distribution of these degrees over the whole network. Scale-free networks contribute to a high degree of error tolerance or resilience; that is, the ability of their nodes to communicate is unaffected by the failure of some randomly chosen nodes.

The lesson here is that the networked nature of the complex adaptive system and its dynamics contribute to the properties of sentience, resilience, and some form of intelligence and is observed even in primitive forms of life.

### 2.1.2. Phase 2: Interacting Networks

With the advent of proteins, networks of proteins have evolved based on their binding properties. In the interacting network model, the proteins are nodes, and two nodes are connected by a non-directed edge if the two proteins bind [26,27]. Physical interactions

dictate the architecture of the cell in terms of how the direct associations between molecules build protein complexes, signal transduction pathways, and other cellular machinery. Genetic interactions define functional relationships between genes, giving insight into how this physical architecture translates into phenotype. Replication, signaling, and composition are essential characteristics required for autopoietic and cognitive processes and they are exhibited in the interactive networks of genes.

### 2.1.3. Phase 3: Neural Networks

To address the questions "How does the brain work?" and "How can we build intelligent machines?" [5], we should look to our understanding from genomics [1], neuroscience [2,3,6], and information science [8–16] all of which are throwing new light. Figure 1 shows our current understanding of how the mind, brain, and body function, based on both theoretical and experimental findings presented by various scholars. On the left-hand side of Figure 1, the genes and neurons are depicted. On the right-hand side, the reptilian complex uses both the brain and body to sense information, convert it into knowledge, and capture cognitive behaviors.

**Figure 1.** Two views of the mind, brain, and body components.

The five senses are used by the reptilian cortical columns to provide embedded, embodied, extended, and enactive (4E) cognitive behaviors. The reference frames [6] provide a knowledge representation in the form of neural networks where the nodes that are fired together based on the inputs from the cortical columns wire together and the nodes that are wired together fire together to exhibit the behaviors using the body to interact with the environment. The knowledge representation captures the relationships between various entities that constitute the system and the entities that the system interacts with. The relationships, along with their dynamic behaviors, are represented in the form of neural networks. The networks of genes provide the autopoietic behaviors using the various physical structures that constitute the body and the brain. The neural networks provide the mechanism for sensing and converting the information into a common knowledge representation across the five senses using the old reptilian brain. The new brain provides higher-level information processing using the common knowledge representation to enhance the 4E cognition from the five senses with the sixth sense of elevated cognition. It is important to note that the genes use physical and chemical processes, converting matter and energy to transform components (exhibiting autopoietic behaviors) using symbolic computing that involves DNA and RNA, where the symbols are characterized by the

4 elements of DNA. The information in DNA is stored as a code made up of four chemical bases: adenine (A), guanine (G), cytosine (C), and thymine (T). Human DNA consists of about 3 billion bases, and more than 99 percent of those bases are the same in all people.

On the other hand, in the neural networks in the brain, a typical neuron collects signals from others through a host of fine structures called dendrites. The neuron sends out spikes of electrical activity through the axon (the output and conducting structure) which can split into thousands of branches. The signals excite the neurons and the neurons that fire together wire together to provide the mechanism for converting information into knowledge, which in turn is used in elevated cognition and in exhibiting cognitive behaviors. This is known as sub-symbolic computing. While the reptilian brain provides sub-symbolic computing-based information, the neocortex uses it to process higher-level functions and uses the network of genes to make appropriate actions to maintain stability within the system and interact with the outside based on its knowledge maintained in the neural networks.

Armed with this knowledge, we can now look for a model to capture the mind, brain, and body, and their relationships to matter, energy, information, and knowledge that enable autopoietic and cognitive behaviors.

### 2.2. Human Intelligence, the Mind-Body Debate and the Theory of Structural Reality

Since the time of ancient Greek philosopher Plato (420s-340s BCE) [28], the material-ideal problem which asks the question—what is the relationship between material things and ideal entities? is the subject of debate even today [10,29–31]. With the new scientific interpretation of Plato's Ideas/Forms [10] in the form of physical and mental structures supplemented by various findings in neuroscience and genomics, we have a new insight into how the mind and body interact. Plato introduced the Theory of Forms. In simple terms, Plato's Theory of Forms asserts that the physical world is not the 'real' world; instead, ultimate reality exists beyond our physical world. The physical world is the material stuff we see and interact with daily; this physical world is always changing and imperfect. The world of Ideas and Forms, however, exists beyond the physical world and exists in the form of ideal structures as demonstrated in [10].

The general theory of information (GTI) [8] and the theory of structural machines [9] dealing with the transformation of information and knowledge tell us that information per se belongs to the ideal world of structures, which is the scientific realization of the world of Plato—Ideas or Forms. However, it comes to, and functions, in the physical world of living beings by acquiring physical representations and physical carriers. Living beings, and the machines invented by them, work with these carriers and physical representations to access, process, use, and store information. On the other hand, the physical universe is made up of structures that interact with each other using the laws of transformation dealing with matter and energy. These interactions result in more complex physical structures. All physical structures contain information, which can change the states of these structures (the concept of from being to becoming).

However, life forms have evolved to use physical structures in the form of genes and neurons to create mental models of physical structures as abstract structures. Unlike physical structures, mental structures only exist in the abstract models created by living organisms using their physical structures. The mental energy required to create mental structures is derived from their physical structures. GTI provides the means for abstract mental model representations using ideal structures that exist in the ideal world. Abstract structures deal with information and knowledge; knowledge to information is as matter is to energy. It provides the tools to transform information into knowledge and follow the dynamics of these mental structures. These are the tools in the form of fundamental triads, named sets, knowledge structures, cognizing oracles, and structural machines. These tools and their uses are also discussed in various books and publications. In the next section, we summarize key lessons from GTI and present the Burgin-Mikkilineni thesis [11] which provides a model not only to describe the super-symbolic, symbolic,

and sub-symbolic computations in living organisms but also a model [11–15] to infuse autopoietic and cognitive behaviors into digital automata.

### 3. General Theory of Information and the Burgin-Mikkilineni Thesis (BMT)

GTI [8–10] provides a "unified context for existing directions in information studies, making it possible to elaborate on a comprehensive definition of information; explain relations between information, data, and knowledge; and demonstrate how different mathematical models of information and information processes are related [32] p. 1." We briefly summarize the tools that GTI provides to model information processing structures and their behaviors both in humans and digital machines.

All material structures contain information and living organisms have developed physical structures (networks of genes and neurons) which gather the information through their senses and convert it into knowledge. The knowledge according to GTI consists of a fundamental triad or a named set [33]. "Named sets as the most encompassing and fundamental mathematical construction encompass all generalizations of ordinary sets and provide unified foundations for the whole mathematics [8] p. 566." According to GTI, "Any natural phenomenon has the structure of some fundamental triad (FUTRAD) or some system consisting of fundamental triads (FUTRADS). As a consequence, fundamental triads and their systems appear to be the basic objects of cognition, and the theory of fundamental triads helps to attain a new and profound understanding of nature's structure and behavior—with a refreshing and simpler (than before) way to describe it. [34] p. 7."

The fundamental triad represents knowledge about structures as shown in Figure 2.

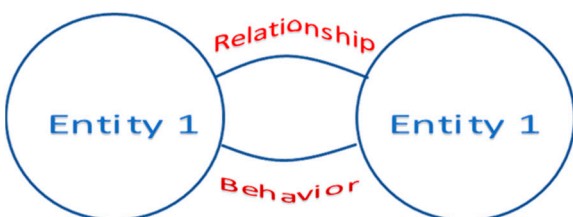

**Figure 2.** The fundamental triad (also known as named set) as a knowledge structure derived from information.

It represents the knowledge in the form of two entities, their relationships, and interactions represented as behaviors. An information unit is described by the existence or non-existence (1 or 0) of an entity or an object that is physically observed or mentally conceived. The difference between an entity and an object is that the entity is an abstract concept with attributes such as a computer with memory and CPU. An object is an instance of an entity with an identity, defined by two components which are the object-state and object-behavior. An attribute is a key-value pair with an identity (name) and a value associated with it. The attribute state is defined by its value. Information is related to knowledge and is defined by the relationships between various entities and their interactions (behaviors) when the values of the attributes change. A named set as a fundamental triad defines the knowledge about two different entities (Figure 2). Each entity, called the knowledge node, receives information through various sensors and transforms it into knowledge based on its internal state which defines various attributes, relationships, and behaviors. Figure 3 shows the structure of a knowledge node.

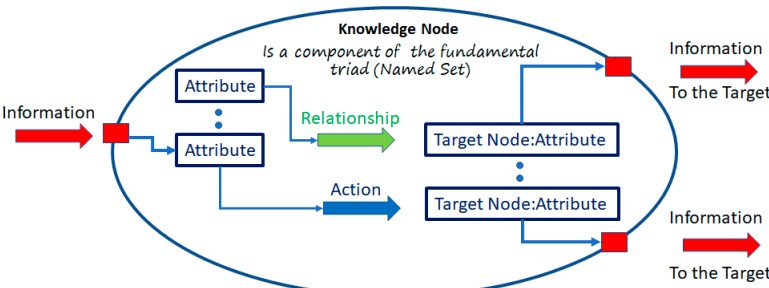

**Figure 3.** Knowledge node represents an entity that receives information and processes it based on its internal knowledge.

A knowledge structure [14] defines various triadic relationships between all the entities that are contained in a system. A knowledge structure is composed of knowledge nodes representing the domain knowledge as a multi-layer complex network depicting various entities, their relationships, and their behaviors.

In essence, a knowledge structure schema and operations provide a process model and its evolution [14]. Figure 4 depicts a knowledge structure as a multilayer network.

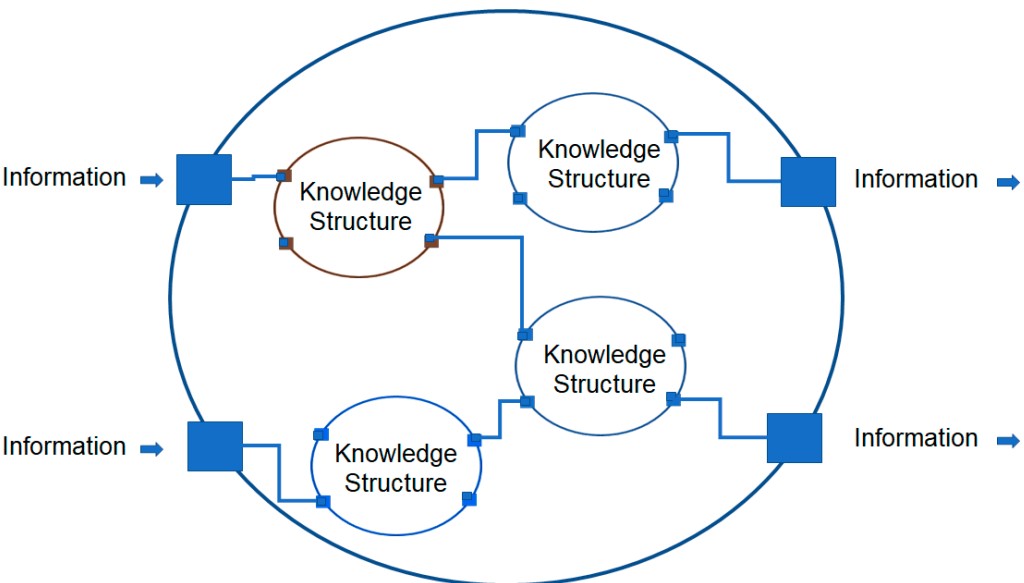

**Figure 4.** A knowledge network consisting of knowledge structures composed of fundamental triads representing knowledge.

As discussed in detail in Burgin, Mikkilineni [14], the knowledge structure provides a common knowledge representation received from various sources using composed fundamental triads. All the knowledge nodes wired together fire together to exhibit collective behavior.

A structural machine [12–14] is an information processing structure that uses the knowledge structures as schema and performs operations on them to evolve information changes in the system from one instant to another when any of the attributes of any of the objects change. The structural machines supersede the Turing machines by their representations of knowledge and the operations that process information [14–17]. Triadic structural machines with multiple general and mission-oriented processors enable autopoietic and cognitive behaviors. The details are discussed on using structural machines, knowledge structure schemas, and operations on them in the Burgin, Mikkilineni paper [14]. The knowledge nodes (shown in Figure 4) are executed by the structural machine using vari-

ous processors with conventional resources. Structural machines operate on knowledge structures in contrast to Turing machines operating on symbolic data structures.

The ontological Burgin-Mikkilineni thesis states that "autopoietic and cognitive behavior of artificial systems must function on three levels of information processing systems and be based on triadic automata. The axiological BM thesis states that efficient autopoietic and cognitive behavior has to employ structural machines. [11] p. 1."

A genome in the language of GTI [11] encapsulates "knowledge structures" coded in the form of DNA and executed using the "structural machines" in the form of genes and neurons which use physical and chemical processes (dealing with the conversion of matter and energy). The information accumulated through biological evolution is encoded into knowledge to create the genome which contains the knowledge network defining the function, structure, and autopoietic and cognitive processes to build and evolve the system while managing both deterministic and non-deterministic fluctuations in the interactions among internal components or their interactions with the environment.

A digital genome [11] is defined as a collection of "knowledge structures" coded in an executable form to be processed with "structural machines" implemented using digital genes (in the form of symbolic computing algorithms) and digital neurons (in the form of sub-symbolic neural net algorithms) both of which use stored program control implementation of Turing machines. The digital genome enables digital process execution to discover the computing resources in the environment, use them to assemble the hardware, cognitive apparatuses in the form of digital genes and digital neurons, and evolve the process of sentient, resilient, intelligent, and efficient management of both the self and the environment with 7e cognitive processes.

The digital genome incorporates the knowledge in the form of multi-layer intelligence with a definition of the sentient digital computing structures that discover, monitor, and evolve both the self and the interactions with each other, and the environment based on best practices infused in them.

The digital genome specifies the execution of knowledge networks using both symbolic computing and sub-symbolic computing structures. The knowledge network consists of a super-symbolic network of symbolic and sub-symbolic networks executing the functions defined in their components [15]. The structure provides the system behavior and evolution maintaining the system's stability in the face of fluctuations in both internal and external interactions. The digital genome encapsulates both autopoietic and cognitive behaviors of digital information processing structures capable of sentience, resilience, and intelligence. The digital genome typifies infused cognition as opposed to evolved cognition in biological systems. The infusion is made by the human operators who teach the machines how to evolve.

In the next section, we will describe how to design a new class of autopoietic and cognitive machines using existing information technologies such as cloud computing, containers, and their management tools just as the neocortex overlay utilized existing reptilian cognitive behaviors.

## 4. Infusing Autopoietic and Cognitive Behaviors into Digital Automata

GTI and structural theories of reality tell us that the material world is composed of structures that deal with transformations of matter and energy. The mental world exists in living beings and is composed of structures that deal with information and knowledge. The mental structures are formed using the physical structures to receive information from various senses, process it to create knowledge structures, and use them to manage the stability, safety, and sustenance of the system. The physical structures used to process information consist of symbolic (networks of genes) and sub-symbolic computing structures (neural networks). The digital world is composed of symbolic and sub-symbolic structures that process information received in the form of symbols and convert it into the knowledge of the state of the system and its evolution.

Symbolic and sub-symbolic computing structures with various algorithms that operate on symbolic data structures have provided significant benefits. These include business

process automation, real-time communication, collaboration, and commerce, etc. Deep learning has delivered a variety of practical uses by revolutionizing customer experience, machine translation, language recognition, autonomous vehicles, computer vision, text generation, speech understanding, and a multitude of other AI applications. However, current symbolic and sub-symbolic computing structures operate as silos and there is no common knowledge representation that brings together the knowledge from these silos together. We propose to use knowledge structures to integrate the knowledge from the silos and provide super-symbolic computing that operates on knowledge structures in contrast to the data structures.

Figure 5 shows a new class of machines that integrate symbolic and sub-symbolic computing structures. Symbolic computing (using algorithms and operations on symbolic data structures) provides the equivalent of the networks of genes that interact with the physical resources using the transformation laws of matter and energy. The neural network algorithms provide the equivalent of extracting information and converting it into knowledge with 4E cognition. Super-symbolic computing provides the mechanism to represent knowledge from multiple sources as a knowledge network. The nodes contain the knowledge structures representing the state of various entities, relationships, and behaviors as an always-on executable service module. The inputs to the knowledge node provide the information that triggers the behavioral changes in the nodes that impact other node behaviors through the communication of information as outputs from the nodes. All the knowledge nodes wired together fire together to execute a collective behavior.

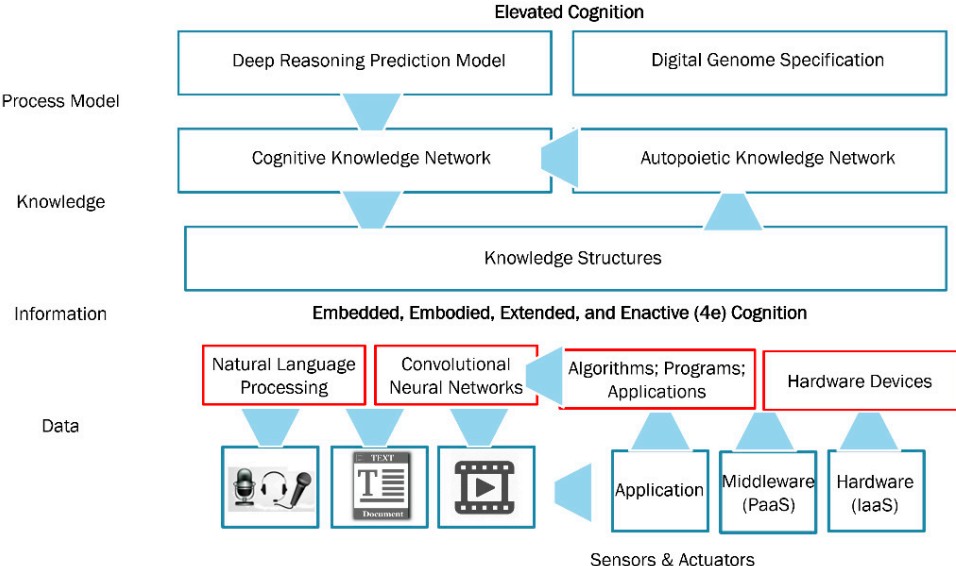

**Figure 5.** Super-symbolic computing structure with a common knowledge representation using knowledge structures and structural machines.

All the knowledge nodes wired together fire together to execute a collective behavior. The behavior is defined by the pre-condition and post-condition constraints. The knowledge network is implemented as a structural machine that provides operations on the Knowledge structure schema [12–14].

A design of the implementation of a structural machine using knowledge structures that represent the "life processes" of a computational workflow of a business software application is depicted in Figure 5. The digital genome shown on the right-hand side is like a cell that contains all the executables, their control structures, and operational details about an application designed to be a distributed computing structure executed in infrastructures offered by different providers. It contains knowledge about what resources (CPU, Memory, storage, network, etc.) are required for each component, where they are available, and how to use them. It is, in essence, a specification of all "life processes" defined in the genome as

knowledge structures (entities, relationships, and behaviors). When the digital genome is executed in a computer as a program to create a specific instance of the business application, it functions as a manager of downstream networks that will be created, monitored, and managed based on the "life process" definitions for these downstream networks. It is done by creating downstream network managers which know what their downstream functions are (like specialized functional cells such as a web server, application server, and a database) that execute specific functions and communicate with other cells influencing their behaviors. All specialized functions are defined using knowledge structures (again entities, relationships, and behaviors executed using local CPU, Memory, OS, Database, file system, program executables, etc.).

The figure shows a digital genome node using cloud resources. It deploys the autopoietic and cognitive knowledge networks. Each leaf knowledge node, in turn, configures, executes, monitors, and manages the various functional tasks that contribute to a knowledge network behavior/. The knowledge functional nodes wired together fire together to execute the collective behaviors. The autopoietic and cognitive behaviors that model the life processes are executed at each layer to maintain global stability and intended outcomes. Deviations are monitored at each level and corrections are made based on life process definitions at each level.

The figure also shows how neural networks and sub-symbolic computing are used to create common knowledge representation from multiple modules executing different algorithms. Knowledge structures and knowledge networks provide a method to create a common knowledge representation from information obtained from multiple means.

The introduction of common knowledge representation in the form of knowledge structures that combine the knowledge obtained from symbolic and sub-symbolic computations in the functional nodes is new. The structural machine implementation with triadic automata [12–14] provides global stability and successful global outcomes from various components to meet the system's goals. while maintaining the local autonomy of individual component management. The super-symbolic overlay manages global optimization while dealing with the fluctuations in the interactions of components impacted by local constraints. The real-time global monitoring and management provide the capability for optimizing global behavior using downstream knowledge network reconfiguration. To the author's knowledge, this approach is the first of its kind in introducing autopoietic and cognitive behaviors in the discussion of digital automata and the path towards strong AI. GTI, structural theory of reality, and the derived tools provide a powerful framework not only to understand the material world but also to design and build a new class of digital machines with improved sentience, resilience, and intelligence. The beauty of this approach is that it utilizes current generation symbolic and sub-symbolic computing structures without disrupting the status quo to create a new class of machines. This is precisely how the mammalian neocortex utilized the classical reptilian cortical columns to integrate knowledge obtained from multiple senses.

## 5. Conclusions

According to Signorelli [35], any attempt to build conscious machines and try to introduce human capabilities should start with the definitions of autonomy, reproduction, and consciousness. In his paper, Signorelli provides a thoughtful discussion about the current state of the art of our understanding from biology, neuroscience, and studies of cognition. He argues that current theories are descriptive in nature and emphasize the need for an explanatory theory that includes causal mechanisms. Any theory should explain autonomy, reproduction, and some form of consciousness to not only understand how the living organisms exhibit these behaviors but also pave the path to design machines that could reproduce these behaviors in some form.

In this paper, we use GTI and the theory of structural reality to provide a model for two unique features, namely autopoiesis and cognition, which are essential in exhibiting the properties of autonomy, reproduction, and consciousness. Autopoiesis requires the

concept of "self" defined as a system composed of autonomous functions having the knowledge to interact with each other and their environment with knowledge of the systemic goal. Cognition requires the functions with the knowledge to receive information from various sources and process it into more knowledge that enables the system to manage its goals with respect to stability, safety, sustenance, and survival in the face of fluctuations in its interactions. Thus, the tools that assist in transforming information and knowledge and manage the functions, structure, and fluctuations are essential to exhibit autonomy, reproduction, and consciousness. Genes and neurons have evolved through natural selection as structures that receive information, process it into knowledge with a common representation, and utilize the knowledge to manage the systemic goals. GTI provides a model to capture these behaviors as fundamental triads and their 'composed' knowledge structures [11].

In addition, GTI paves the path to model these behaviors using the digital genes and digital neurons in the digital world, and paves the path for infusing autopoiesis and cognition with varying degrees in digital automata. It also points out that the structural machines and cognizing oracles are required to go past the limitations of the Turing machine computing model to introduce the sense of self and systemic goals and manage the functions, structure, and fluctuations using information-processing structures.

Therefore, this paper brings together our learning from genomics, neuroscience, and the science of information processing structures to show a new path to design and build autopoietic and cognitive machines. These machines go beyond the current state of the art in mimicking living organisms. GTI integrates the knowledge gained from various philosophers, mathematicians, and eminent thinkers from Plato to Burgin and suggests a possible new path to classical computer scientists and IT professionals who are pursuing symbolic and sub-symbolic computing structures to mimic living beings. GTI and the theories of structural reality, while not widely recognized by mainstream scholars, provide a unified framework for understanding material structures and mental structures, reason about them, and relate them to ideal structures described by Plato's ideas/Forms. As Burgin points out, they also provide a framework to discover the relationship between computing, communication, cognition, consciousness, and culture (the five Cs) that permeate the material and mental worlds [11]. This paper is an attempt to bring these concepts to build a new digital world. As far as the related work is concerned, we believe that the application of GTI to model autopoietic and cognitive behaviors is novel and the author is not aware of any similar proposal except for the references provided in this paper.

**Funding:** This research received no external funding.

**Institutional Review Board Statement:** Not applicable.

**Informed Consent Statement:** Not applicable.

**Data Availability Statement:** Not applicable.

**Acknowledgments:** The author wishes to thank Mark Burgin for various in-depth discussions on the general theory of information and the theory of structural machines. Thanks also to Gordana Dodig Crnkovic for introducing the author to various aspects of cognition and many valuable discussions. This paper is mainly a result of these discussions. The author wishes to express his gratitude to the late Peter Wegner for many discussions that shaped his understanding of computer science. The author also expresses his gratitude for the education he received at the University of California, San Diego where many Nobel Laureates and other eminent scholars influenced his thinking. Walter Kohn (Nobel Laureate 1998) had a special influence as a thesis advisor and mentor.

**Conflicts of Interest:** The author declares no conflict of interest.

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
