# Peer review of "A New Class of Autopoietic and Cognitive Machines"

_information, doi:10.3390/info13010024_

Round 1
Reviewer 1 Report
The work proposes a path to extract knowledge from cognitive science theories, especially proposing to map the observed and theorized behavior of humans' brains into machine-based systems.
The paper addresses a captivating research path: how to allow machines to have characteristics of self-adaptation and generalization over the environment.
The topic is exciting, and some proposals were made in the paper, like the "hierarchical" approach for addressing the inputs information and trigger systems' changes.
However, the writing seems complex and intricate, and the connection between different sections is missing. It is difficult for the reader to grasp all the concepts, connect them, and relate them to a gap that is not readily identifiable in the text. In some parts, especially in the introduction, the author introduces many concepts without accompanying them with citations. Furthermore, other works are cited blatantly, without really going in-depth with what these approaches offer and how the concepts can be mapped into a distributed system *aaS scenario (which appears only in later sections).
Therefore, it would be good if the author would take some time to restructure the paper and bring it to new life with a new form.
Author Response
Thank you for your comments:
However, the writing seems complex and intricate, and the connection between different sections is missing. It is difficult for the reader to grasp all the concepts, connect them, and relate them to a gap that is not readily identifiable in the text. In some parts, especially in the introduction, the author introduces many concepts without accompanying them with citations. Furthermore, other works are cited blatantly, without really going in-depth with what these approaches offer and how the concepts can be mapped into a distributed system *aaS scenario (which appears only in later sections).
Therefore, it would be good if the author would take some time to restructure the paper and bring it to new life with a new form.
I have restructured and revised the paper and added more references and details.
Reviewer 2 Report
An extraordinary article that builds close links among neuroscience, genetics, AI and information science. I would assign to a strong seminar in Cognitive Science. It combines medical, biological knowledge not only with AI but also with very strong philosophical perspective on Information and related domains. A few infelicities have been marked in the version of the paper with my remarks. The Author should define his terms a bit more clearly, especially 'autopoiesis' but also several terms from M. Burgin's works. Also, some aspects of neuroscience/genetics are a bit compressed. The author should not expect the readers to be as versed as he is in the unusual mix of all those competency areas. Moreover, the article is not excessively long; thus, going a bit more slowly through intricate topics would be a great benefit.
This reviewer also noted unusual use of some of the references (if they include page numbers, they should rather go after the actual quote). One of the tables is unreadable since it is the size of a poster presentation -- this should be fixed -- maybe by replacing it with a series of smaller graphs that can be reproduced legibly. Finally, the material in the abstract does not count as a part of the presentation -- therefore all pieces of it must be repeated, and developed, in the actual body of the article.
A one easy rewrite should take care of all those infelicities.
The enclosed file is the text of the manuscript with some comments.

Author Response
I am grateful for the very thoughtful and detailed suggestions to improve the paper. I have rewritten the paper and incorporated all suggestions made by the reviewer.
Reviewer 3 Report
With the emergence of deep learning, the Fourth Industrial Revolution comes.
Professor Burgin's paper "Triadic Automata and Machines as Information
Transformers" is published in this information journal of mdpi, since this paper
shows new aspect of triadic automata system as a big mathematical scholar.
This new paper of Professor Rao Mikkilineni extends Burgin's theory in section 3 and 4.
His new view of neural network is shown in section 2. I think that this paper must
deal with new technology like CNN (Convolutional Neural Network) and transformer.
Only multi-layer perceptron is described in this theory. In this reason, the content
of this paper is a somewhat routine. It is too boring to read, although this paper
contains an interesting aspect.
Specific comments.
* I think that section 3 and 4 must be removed to simplify
this paper.
* The section 2 must contain the adaptation to CNN and transformer of
deep learning.
* The application of triadic automata to cloud computing is not clear.
Triadic automata must be applied directly to new emerging deep learning network.
Author Response
Thank you for the comments and suggestions. I revised and rewrote the paper following all the suggestions from the three reviewers. The responses to this reviewer's comments are in line in red:
With the emergence of deep learning, the Fourth Industrial Revolution comes.
Having participated in the AI revolution since the 1980's and going through the various AI winters, the author is skeptical of deep learning as the sole reason for 4th industrial revolution. Strong AI mimicking living organisms has been the holy grail of computer science and has eluded the classical computer scientists and information technology professionals so far. While the word cognition has been associated with deep learning, it may not be the whole story as Prof. Mark Burgin's writings show. Unfortunately, classical computer scientists have completely missed these profound writings which, in the author's view are as important as special theory of relativity (which generalized Newtonian classical physics) and statistical mechanics (which provided a microscopic view of macroscopic behaviors with function, structure and fluctuations).
Professor Burgin's paper "Triadic Automata and Machines as Information
Transformers" is published in this information journal of mdpi, since this paper
shows new aspect of triadic automata system as a big mathematical scholar.
This new paper of Professor Rao Mikkilineni extends Burgin's theory in section 3 and 4. His new view of neural network is shown in section 2. I think that this paper must
deal with new technology like CNN (Convolutional Neural Network) and transformer.
Only multi-layer perceptron is described in this theory. In this reason, the content
of this paper is a somewhat routine. It is too boring to read, although this paper
contains an interesting aspect.
The discussion is more general than incrementally improving CNN or RNN. The focus of this paper is to identify the key ingredients of living organisms that give them the unique properties of sentience, resilience and intelligence. I believe that Prof. Burgin's writings on general theory of information and the structural theories of reality provide new insight into how to model the key ingredients (autopoiesis and cognition) and also allow us to infuse them into current state of the art including both symbolic and deep learning. The super-symbolic computing proposed here uses both symbolic and deep learning networks to create a new coomon knowledge representation and do deep reasoning just as the mammalian neocortex repurposed the reptilian cortical columns that provide information processing from the five senses (See Jeff Hawkins, (2021). “A Thousand Brains: A New Theory of Intelligence.” Basic Books, New York.)
Specific comments.
* I think that section 3 and 4 must be removed to simplify
this paper.
* The section 2 must contain the adaptation to CNN and transformer of
deep learning.
* The application of triadic automata to cloud computing is not clear.
Restructured and revised the paper following all three reviewer's suggestions and addresses these specific comments.
Round 2
Reviewer 1 Report
May I suggest that you reconsider citing Plato and then writing "Professor Burgin" - this sounds irritating. With no other citation one writes the academic titles, hence refraining from this one, also appears to be appropriate.
Author Response
May I suggest that you reconsider citing Plato and then writing "Professor Burgin" - this sounds irritating.
Added Plato citation and revised the paper
Ran spell check, Grammarly, and Turnitin. Made appropriate revisions.
Thank you
Reviewer 3 Report
Thank you for your revision.
Many changes were performed in this revision. Neural network concepts are introduced in several locations.
I think that since Burgin's theory has an weak foundation in my view, an extension of his theory in sci journal is too early.
Following paper is the good paper in my aspect.
"Signorelli CM (2018) Can Computers Become Conscious and Overcome Humans? Front. Robot. AI 5:121. doi: 10.3389/frobt.2018.00121"
This paper is easily understandable and has many citations.
Under the weak theory, construction of new concepts is problematic, therefore readers must find other reference papers of Burgin to
get meaningful information from this paper.
I recommend to find the concept of Yi Hwang known as one of the most reversed neo-confucian scholars of the joseon period(1392-1910).
My grandmother is a direct ancestor of Yi Hwang. She lived in Yangdong village with her parents.
This village is a historic one of korea selected by unesco.
You can find his profile in this homepage.
(http://dh.aks.ac.kr/Korea100/wiki/index.php/Renowned_Neo-Confucian_Scholar,_Yi_Hwang)
I recommend that your concept can be derived from neo-confucian. Although neo-confucian is not my major, most south koreans learn abstract concepts
of ri-gi theory from mother tongues.
(you can find his theory in this location, and attach this pdf file.: https://www.kci.go.kr/kciportal/ci/sereArticleSearch/ciSereArtiView.kci?sereArticleSearchBean.artiId=ART001851048)

Author Response
Thank you for your thoughtful comments and suggestions. I have tried to incorporate as much as possible and here are my responses inline:
1. I think that since Burgin's theory has a weak foundation in my view, an extension of his theory in sci journal is too early.
Following paper is the good paper in my aspect.
"Signorelli CM (2018) Can Computers Become Conscious and Overcome Humans? Front. Robot. AI 5:121. doi: 10.3389/frobt.2018.00121"
This paper is easily understandable and has many citations.
I am grateful for introducing this paper to me. I have not only referenced it but also included some of his arguments in my revised conclusion and attempted to relate his work with the content of this paper.
2. Under the weak theory, construction of new concepts is problematic, therefore readers must find other reference papers of Burgin to
get meaningful information from this paper.
I agree with you that it is hard to apply GTI from the writings of Burgin directly. However, his theory is about mathematical and philosophical aspects showing the shortcomings of the Turing computing model and suggesting the structural machine approach, which is shown to be more efficient, resilient, and allows new capabilities. It is more like Maxwell’s theory which was later used to discover radio waves and invent new applications. My paper is an attempt to show how GTI can be applied to build knowledge structures and structural machines using current state-of-the-art symbolic and neural-network programs. In order to build them, one need not know the mathematics behind them. Already, I know a couple of groups in the USA are building such software systems applying these ideas. The first one is to provide self-managing application workloads that can auto-scale, auto-failover, and self-migrate across clouds using local IaaS and PaaS resources. The second one is a super-symbolic structure that integrates existing neural network algorithms and symbolic computing programs with cognitive overlay. This allows the capability to change the course of a function downstream with global knowledge based on the results from another function that directly does not have any relationship with the function that it is impacting. This allows us to build real-time global changes from information from locally autonomous functions that interact with their environment which is globally connected.
3. I recommend to find the concept of Yi Hwang known as one of the most reversed neo-confucian scholars of the joseon period(1392-1910).
My grandmother is a direct ancestor of Yi Hwang. She lived in Yangdong village with her parents.
This village is a historic one of korea selected by unesco.
You can find his profile in this homepage.
(http://dh.aks.ac.kr/Korea100/wiki/index.php/Renowned_Neo-Confucian_Scholar,_Yi_Hwang)
I recommend that your concept can be derived from neo-confucian. Although neo-confucian is not my major, most south koreans learn abstract concepts
of ri-gi theory from mother tongues.
(you can find his theory in this location, and attach this pdf file.: https://www.kci.go.kr/kciportal/ci/sereArticleSearch/ciSereArtiView.kci?sereArticleSearchBean.artiId=ART001851048)
Thank you for introducing the neo-Confucian theory and I am going to study it in-depth and compare other eastern approaches with the western approaches. This is a very interesting philosophical exercise to see how Indian writings, the zen approach of Japan, and the neo-Confucian approach deal with information, knowledge, and consciousness. What are their views about physical reality, mental reality, and spiritual reality and their connections? This would be a fascinating study and requires a lot of work, I think I will enjoy it
Again, thank you for your patience.
Round 3
Reviewer 3 Report
This paper is much improved in the aspect of easy reading. In the western philosophy, the theory of this paper shows many interesting points. After this research is published in a journal, much discussion will be made from other research groups, because this paper includes many challenging themes. If the comparison with eastern philosophy is included, the discussion of this paper will be more improved.